# Cost-effective DNA extraction method optimized for high yield and long fragments from coastal sediments

**Leopold Matthys**[1], **Eléonord Mayissah Moungues**[1,2], **Anaëlle Genève**[1],
**Virginie Sanial**[1], **Benjamin Misson**[1], **Elisabeth Navarro**[1]*

**1** Université de Toulon, Aix Marseille Univ., CNRS, IRD, MIO, Toulon, France, **2** Laboratoire de Recherche Multidisciplinaire en Environnement (LARME), Université des Sciences et Techniques de Masuku, Franceville, Gabon

* elisabeth.navarro@ird.fr

## Abstract

Efficient DNA extraction from coastal sediments is a critical step for studying microbial communities, especially when targeting high-molecular-weight DNA for long-read sequencing. Here, we compare a cost-effective laboratory-made (LM) extraction method with two widely used commercial kits: Qiagen DNEasy™ PowerSoil™ Pro (PSP) and PowerMax™ Soil (PM). In sandy, low-organic-matter sediments, the LM method consistently produced higher DNA yields – up to 13 times more than the commercial kits – and longer DNA fragments (up to 4-fold), likely due to its enzymatic lysis approach. It was also the only method that successfully amplified the 18S rRNA gene in sufficient quantity to enable downstream sequencing in certain samples. While prokaryotic diversity metrics were largely similar across methods, we observed differences in eukaryotic richness that were likely influenced by the amount of sediment processed. This demonstrates the higher importance of sample amount when assessing microeukaryotic diversity. Despite these differences, the LM method gave reproducible results and recovered taxa that were less frequently detected by commercial kits. In terms of cost, the LM method is significantly cheaper per sample than commercial alternatives, despite a higher starting cost. Although it requires more hands-on time, this trade-off may be worthwhile for large-scale studies and developing countries. Overall, the LM method is a robust and economical option, particularly suited for sandy or low-biomass sediments, and for studies requiring high-quality DNA. These results underline the importance of choosing and validating extraction protocols based on both the sediment type and the study's goals.

## Introduction

Coastal microorganisms are important actors in the functioning of land-sea transition zones, where they drive critical ecosystem functions such as nutrient cycling,

**Data availability statement:** All relevant data has been deposited on Zenodo at the following DOI: https://doi.org/10.5281/zenodo.16083979. MiSeq reads were deposited in the National Center for Biotechnology Information Sequence Read Archive (NCBI SRA) under the accession number PRJNA1255512.

**Funding:** This research was funded by a MESR doctoral contract from the French Ministry of Higher Education and Research (Ministère de l'Éducation nationale, de l'Enseignement supérieur et de la Recherche), awarded to LM at Université de Toulon. This work was supported by the French national program EC2CO (Ecosphère Continentale et Côtière) under the BRONZETTE2.0 project. The funders had no role in study design, data collection and analysis, decision to publish, or preparation of the manuscript.

**Competing interests:** The authors have declared that no competing interests exist.

organic matter degradation and primary production [1–3]. Coastal sediments harbor highly abundant and diverse (in both taxonomy and function) microbial communities that help maintaining a healthy ecosystem [1,4–7]. For the past 15 years, the use of metabarcoding and metagenomics has transformed the study of microbial communities, enabling taxonomic and functional characterization of complex communities without cultivation [8,9]. Advances in technology that allow next-generation sequencing (NGS), and more recently long-read sequencing, now grants deeper and more resolved insights in environmental and bacterial DNA at affordable cost [10]. Metabarcoding targets specific genetic loci (e.g., fragments of 16S or 18S rRNA genes) to generate taxonomic profiles from established marker genes with high coverage of the genetic diversity for the targeted loci. In contrast, shotgun metagenomics sequences all DNA present in a sample, providing access to wider range of taxonomic and functional information, while providing less sequencing depth on each sequenced genomic region. Third-generation sequencing further improves shotgun metagenomics by producing long, continuous reads that help resolve complex regions and strengthen genome reconstruction. It also enables long-read metabarcoding, allowing full-length marker sequencing and improved taxonomic resolution. However, those technologies are still dependent on DNA extraction methods, indicating the importance of the efficiency of DNA extractions to recover high yields of high-quality DNA (long fragments and inhibitor free) and capture broader microbial diversity. DNA extraction from sediments faces specific challenges, linked to the high variability in biomass [11,12] and enzymatic interference by the co-extraction of inhibitors [13,14]. Available non-automated methods are commercial kits that may have low yields and extract poor-quality DNA [15,16], or in-house protocols that usually require hazardous reagents (e.g., phenol and chloroform purification steps [16,17]). The goal of this study was to propose a cost-effective, less hazardous, laboratory-made (LM) method, optimized for high yield and quality DNA. To assess the efficiency of the LM method, DNA extraction yield, fragment size, alpha diversity (within-sample taxonomic richness and evenness) and beta diversity (between-sample compositional dissimilarity) from 16S rRNA and 18S rRNA sequencing were compared to two commercial kits on coastal sediments of different origin, texture, and organic matter content.

## Methods

### Sediment collection and characterization

Four distinct sediments with different origin, characteristics and organic matter content were studied (characteristics, location, organic and water content are described in Table 1). While no permits were required to access the publicly accessible sampling sites in France, sampling in Gabon was conducted under authorization from the Centre National de la Recherche Scientifique et Technologique (CENAREST). One mangrove sediment sample (hereafter MANG) was collected in a coastal area (near Alembétogo, Gabon) using a soil auger at a depth of 5 centimeters. One harbor sediment sample (hereafter HARB) was retrieved from a small marina (Oursinieres harbor, Le Pradet, France) using an Ekman grab. Two sandy sediments were collected

**Table 1. Sediment characteristics.**

| Sediment designation | MANG | HARB | EstFS | EstCS |
|---|---|---|---|---|
| Origin | Mangrove | Harbor | Subterranean estuary | Subterranean estuary |
| Characteristics | Mud | Mud | Fine sand | Coarse sand |
| Location | Alembétogo, Gabon | Le Pradet, France | La Londe-les-Maures, France | La Londe-les-Maures, France |
| Latitude | 1.296139° S | 43.084742° N | 43.121325° N | 43.121293° N |
| Longitude | 9.033556° E | 6.018672° E | 6.270798° E | 6.270794° E |
| Total carbon (mg g$^{-1}$) | 15 | 80 | 0.095 | 0.19 |
| Total nitrogen (mg g$^{-1}$) | 0.92 | 4.3 | 0.12 | 0.14 |
| C/N | 16 | 19 | 0.80 | 1.3 |
| Water content (%) | 21 | 62 | 20 | 13 |

from a subterranean estuary located on Pellegrin Beach (La Londe-les-Maures, France). Upstream, approximately 6 meters from the sea, a hole was dug until the water table was visible (~ 0.2-meter depth) and sediments composed of fine sand (hereafter EstFS) were collected on top of the water table. Downstream of the subterranean estuary, one meter from the sea, coarse sand (hereafter EstCS) was collected the same way. At each site, sediments were collected in three independent 50 mL Corning tubes to capture within-site variability. Those aliquots were kept in the dark and frozen at −20°C upon arrival at the laboratory (within a few hours), except MANG that was kept at 4°C for two weeks before freezing at −20°C. Water content of each sample was calculated by weighing the sediment before and after incubation at 60°C for at least 48 hours. Total carbon and nitrogen were measured on dry sediments using an elemental analyzer (Flash 2000 Organic Elemental Analyzer, Thermo Fisher Scientific). Quality of measurements was controlled with a NC soil standard (ref 338 40025, Elemental Microanalysis) with a detection limit of 3 μg of C and 0.5 μg of N in 35 mg of material. For each DNA extraction method, three separate DNA extractions were performed on three thawed biological triplicates for each sediment.

## Nucleic acids extraction

**The proposed laboratory-made method (LM): CTAB and SDS-based enzymatic lysis method with linear polyacrylamide precipitation.** This method is based on enzymatic cell lysis, digestion and precipitation of proteins, prior to DNA recovery by precipitation with isopropanol aided by linear polyacrylamide (GenElute™, Sigma-Aldrich). It combines cell lysis and protein digestion steps inspired from a seawater DNA extraction method [18] and lysis buffer composition of Zhou et al. (1996) [17]. In comparison to Zhou et al. (1996) [17], hexadecyltrimethylammonium bromide (CTAB) and sodium dodecyl sulphate (SDS) additions were delayed in the lysis steps, phenol chloroform purification was changed for Protein Precipitation Solution (PPS) and CTAB additions, and precipitation was aided by linear polyacrylamide. Also, LM described here uses only 200 mg of starting material in comparison to a few grams of soil and sediment in other methods [15,17,19]. To recover more DNA, LM can be adapted for larger amounts of starting material. To prevent increased co-extraction of inhibitors, lysis and precipitation volumes must be increased proportionally to maintain the same sediment-to-reagent ratio. For example, successful extractions were performed using 2 g of sediment in 6.6 mL of lysis buffer and CTAB in 15 mL Corning tubes. To optimize extraction yield and fragment size, different lysis buffer composition, CTAB and SDS concentrations as well as protein precipitation solution (PPS) additions were tested using 200 mg of starting material (S1 File). A simplified workflow of LM is presented in Fig 1. All steps were performed gently, and no vortex was used. This method takes approximately 8 h to extract DNA of 24 samples.

Going into details, the lysis buffer is composed of 100 mM Tris-HCl (pH 8, Fisher, France), 10 mM ethylenediaminetetraacetic acid (EDTA) (pH 8), 1.5 M sodium chloride, 100 mM sodium phosphate (pH 8). The lysis buffer

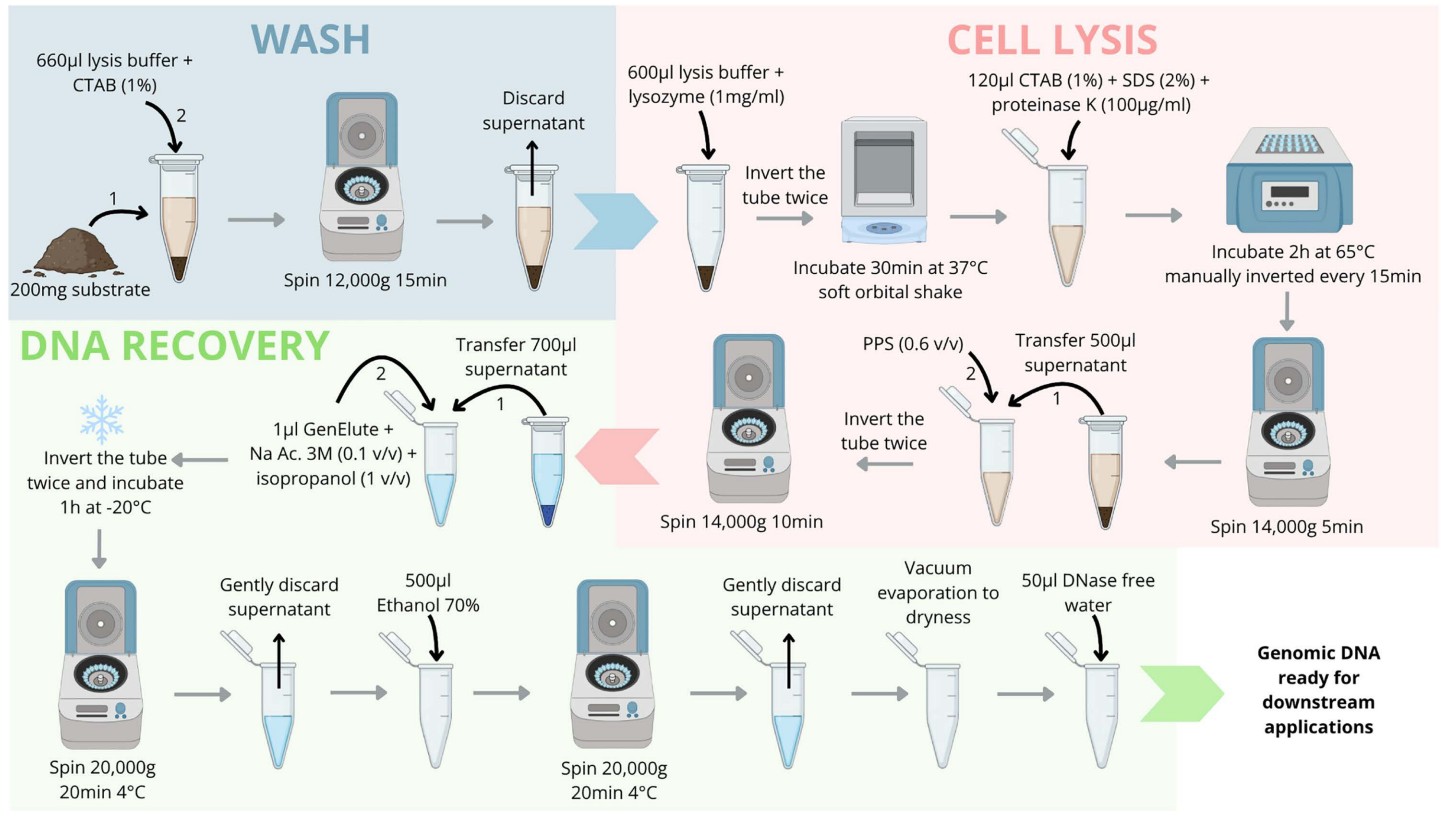

**Fig 1. Synthetic workflow for the LM DNA extraction method.** Concentrations in parentheses are final concentrations. CTAB: hexadecyltrimethylammonium bromide. PPS: Protein Precipitation Solution. SDS: Sodium dodecyl sulfate. Na Ac. 3 M: Sodium acetate 3 M pH 5.2.

was prepared and autoclaved before use. A volume of 660 μL of lysis buffer and CTAB (dissolved in MilliQ water at 1% final concentration, final conc.) was added to 200 mg of sediment into a 2 mL Eppendorf tube. The tube was inverted twice then centrifugated (15 min 12,000 g at room temperature). After discarding the supernatant, the pellet was resuspended in 600 μL of lysis buffer and lysozyme (1 mg mL$^{-1}$ final conc.) and incubated 30 min at 37°C with soft orbital shake (225 rpm, Incu-shaker Mini, Benchmark). After incubation, 120 μL of CTAB, SDS (Fisher, France) and proteinase K dissolved in MilliQ water were added (final conc. of 1%, 2% and 100 μg mL$^{-1}$ respectively), prior to incubation for 2 h at 65°C. During this incubation, tubes were manually inverted every 15 min. After cell lysis, tubes were centrifuged (5 min 14,000 g at room temperature), then 500 μL of supernatant was transferred into a new 2 mL Eppendorf tube. A volume of 0.6:1 (volume, v/v) of PPS (Promega, France) was added. Tubes were inverted twice then centrifuged (10 min 14,000 g at room temperature). A volume of 700 μL of supernatant was transferred into a new 2 mL Eppendorf tube, followed with an addition of 1 μL of GenElute™, sodium acetate (3 M pH 5.2, ThermoFisher, 0.1 v/v) and isopropanol (Fisher, 1 v/v). Tubes were inverted twice and incubated for 1 h at −20°C to allow for nucleic acids precipitation. DNA was recovered by centrifugation (20 min 20,000 g at 4°C) and careful supernatant removal. Pellets were washed with 500 μL ethanol 70% and centrifuged (20 min 20,000 g at 4°C). Most ethanol was removed by discarding the supernatant, the remaining ethanol was evaporated in a vacuum sealed container for ca. 15 min. Dry DNA pellets were solubilized in 50 μL DNase free water. After solubilization for ca. 15 min, DNA were frozen at −20°C until being further analyzed.

**Qiagen DNeasy™ kits: PowerSoil™ Pro (PSP) and PowerMax™ Soil (PM)**

The efficiency of the LM method on various sediments was compared to two commercial kits: DNEasy™ PowerSoil™ Pro (PSP) and DNeasy™ PowerMax™ Soil (PM) (Qiagen, France). The PSP kit is very often used in sediment and soil DNA extraction, known for its high extraction yields in diverse soil types [20]. However, the recommended quantity of starting material limits the amount of DNA recovered, hence why some studies use the PM kit, recommended for low biomass soils. Metrics used to compare the methods include DNA concentration and extraction yield, fragment size, amplification success of taxonomic markers, and diversity via metabarcoding.

Using the PSP kit, 250 mg of sediment were used according to the manufacturer's recommendations. DNA was recovered in 50 µL. With the PSP method, DNA can be extracted from 24 samples in approximately 3 h. For the PM kit, the manufacturer recommends using 5 g of material for high-organic-matter sediments, and up to 10 g for low-organic-matter sediments. Subsequently, extractions were performed using 5 g of MANG and HARB, and 10 g of EstFS and EstCS. DNA was recovered in 5 mL stored in 1 mL aliquots at −20°C for conservation before analysis. With the PM kit, a series of 6 samples requires approximately 4 h.

**DNA quantification and size evaluation**

Double-stranded nucleic acids were quantified using QuantiFluor™ dsDNA System (Promega) on a Tecan Infinite 200 Pro microplate reader following the manufacturer's recommendations. DNA integrity and size distribution ranging from 200 to 60,000 bp was assessed by automated electrophoresis using the Genomic DNA ScreenTape kit on TapeStation 4150 (Agilent Technologies, France). DNA Integrity Number (DIN) and DNA peak table were retrieved from the TapeStation Software.

**PCR amplification and sequencing**

To evaluate prokaryotic diversity, the V4-V5 regions of 16S rRNA gene were amplified using primers 515F-Y and 926R [21] yielding 410 bp amplicons. Eukaryotic diversity was assessed by amplifying the V7 region of 18S rRNA gene using primers 960F and NSR1438 [22] yielding 260 bp amplicons. All primers were coupled with Illumina adaptors for NGS. Amplification mix was composed of 10 ± 2 ng of sample DNA, 2X GoTaq LongMasterMix (Promega, France), 0.4 µM of each primer and DNase free water in a final volume of 30 µL. For 18S amplification, samples that failed to produce sufficient PCR product with 10 ng of starting DNA were re-amplified using an average of 20 ng of DNA. Those samples include MANG and HARB extracted with PM, as well as one replicate of HARB extracted with either PSP or LM. Amplification cycles for prokaryotes were composed of initial heating step of 2 min at 95°C, then 25 cycles of 95°C for 30 s, 50°C for 45 s, 72°C for 45 s and a final extension at 72°C for 10 min. Amplification cycles for eukaryotes were composed of initial heating step of 5 min at 95°C, then 25 cycles of 95°C for 30 s, 53°C for 30 s, 72°C for 30 s and a final extension at 72°C for 7 min. PCR amplification were checked by electrophoresis on 0.7% agarose gel with 1 kb DNA ladder (FastGene 1 kb DNA Marker Plus, NIPPON Genetics). Each DNA sample was amplified once; no technical PCR replicates were performed. PCR controls did not show amplification results on gel. Extraction blanks for PSP and PM did not provide any amplification. For LM, a faint amplification was observed in the extraction blanks; however, the DNA quantity was insufficient for library preparation and sequencing. Consequently, no sequencing data could be obtained from the LM extraction blanks. Amplicons were purified using QIAquick PCR purification kit (Qiagen). Purified amplicons were paired-end sequenced (2 x 250 bp) by Eurofins Genomics with MiSeq Illumina technologies. Amplicons of extractions blanks for LM could not be sequenced because of too low starting material. MiSeq reads were deposited in the National Center for Biotechnology Information Sequence Read Archive (NCBI SRA) under the accession number PRJNA1255512.

**Bioinformatic processing**

Eurofins Genomics provided MiSeq reads that were trimmed of Illumina adaptors using Cutadapt software v2.7 [23]. Reads of 16S and 18S fragments were then separately processed using DADA2 v1.30.0 [24] on RStudio v2024.4.2.764

[25], running on R v4.3.2 [26]. 16S reads were trimmed (240F, 200R) and filtered (maxN = 0, maxEE = c(2,2), truncQ = 2, rm.phix = TRUE, compress = TRUE). 18S reads were trimmed (180F, 150R) and filtered with the same parameters as 16S reads. Sample inference, paired-read merge, sequence table construction and chimeras' removal steps were performed with standard parameters of the DADA2 pipeline [18,24]. For each amplicon sequence variant (ASV), prokaryotic taxonomy was assigned with SILVA v138.2 [27] and eukaryotic taxonomy with PR2 v5.0.0 [28]. Sequences were aligned with DECIPHER v2.30.0 [29] and phylogenetic trees were constructed using FastTree v2.1 [30]. Chloroplasts, Mitochondria and eukaryotic sequences were removed from the prokaryotic dataset. Metazoan and *Embryophyceae* were filtered out of the eukaryotic dataset. Read counts were normalized to relative abundances using Total Sum Scaling (TSS).

## Statistical analyses

All the data are presented as mean ± standard deviation. Univariate differences were assessed using Welch's t-tests performed with the R package rstatix v0.7.2 [31]. Univariate analyses of variance (ANOVA) and post-hoc comparisons (Tukey) were performed with the base stats package. Multivariate analyses of variances were performed with the vegan package v2.6-10 [32]. Alpha diversity indices and phylogenetic distances were calculated with phyloseq v1.46.0 [33]. Discriminant taxa between methods were determined by Linear Discriminant analysis (LDA) effect size (LEfSe) via microbiomeMarker v1.8.0 [34]. LEfSe parameters were the following: norm = "CPM", taxa_rank = "none", kw_cutoff = 0.05, lda_cutoff = 2. Shared and unique ASVs were defined based on presence in all three replicates of a given sediment for each method. Across sediments, shared and unique ASVs between methods were represented in an Upset plot using ComplexUpset v1.3.6 [35]. Enriched ASVs in LM were selected by running two separate LEfSe analyses: one separating all three methods and selecting ASVs specifically enriched in LM; another combining PSP and PM selecting ASVs that were not enriched in the combination. Inversely, depleted ASVs in LM were selected the following way: ASVs enriched in PSP and PM, and ASVs specifically depleted in LM compared to PSP and PM altogether. In all analyses, the p-value threshold for statistical significance was 0.05. Plots were made with ggplot2 v3.5.2 and metacoder v0.3.8 [36,37].

## Results

The efficiency of DNA recovery was assessed by comparing extraction yield and fragment size across all methods (LM, PSP and PM). The influence of the extraction method on microbial communities was also evaluated with alpha diversity metrics, phylogenetic distances and discriminant taxa between methods.

### Efficiency of DNA recovery

**Genomic DNA concentrations and extraction yield.** DNA concentrations were measured for each sediment extracted with methods LM, PSP and PM. Overall, DNA concentrations were significantly different between methods across sediments (ANOVA p = 0.027). With the LM method, concentrations varied across sediments with $14 \pm 4$ ng $\mu L^{-1}$ in MANG, $71 \pm 8$ ng $\mu L^{-1}$ in HARB, $4.7 \pm 1.5$ ng $\mu L^{-1}$ in EstFS and $1.6 \pm 0.5$ ng $\mu L^{-1}$ in EstCS. MANG and HARB sediments extracted with PSP displayed high DNA concentrations ($73 \pm 34$ ng $\mu L^{-1}$, and $70 \pm 9$ ng $\mu L^{-1}$ respectively) but not statistically different to LM (PSP-LM t-test p > 0.05 for both sediments). PSP extracted DNA in significantly lower concentrations than with LM in sandy sediments ($0.69 \pm 0.2$ ng $\mu L^{-1}$, t-test p = 0.032 in EstFS and below $0.026$ ng $\mu L^{-1}$, t-test p = 0.022 in EstCS). With PM, average DNA concentrations ranged from $0.31 \pm 0.08$ ng $\mu L^{-1}$ in EstCS to $7.0 \pm 0.3$ ng $\mu L^{-1}$ in HARB.

Given the differences of initial sediment mass and final volume in DNA extract between methods, the efficiency of DNA recovery can be assessed by comparing the extraction yield. DNA extraction yields were calculated based on DNA concentration, final extract volume and initial sediment mass (dry weight). Overall, LM showed significantly higher extraction yields compared to at least one method (PSP or PM) except for MANG ([Fig 2]). Indeed, in HARB, LM yielded $46 \pm 5$ µg of DNA per gram (dry weight, dw), significantly higher than PSP and PM by 1.3-fold and 2.4-fold, respectively (t-test p = 0.046 and p = 0.006). In EstFS, extraction yield was 13-fold higher with LM than with PSP ($1.4 \pm 0.5$ and $0.16 \pm 0.05$ $\mu g_{DNA}$ $g_{dw}^{-1}$,

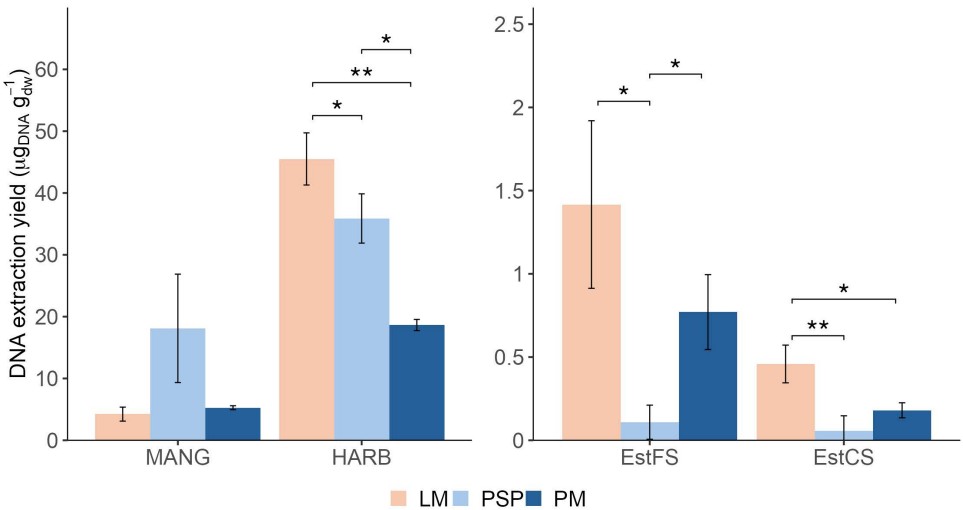

**Fig 2. DNA extraction yield of LM, PSP and PM across sediments.** Yield expressed in µg of DNA extracted per gram of sediment (dry weight, dw). Statistical significance was assessed using a t-test. LM: Laboratory-made method. PSP: PowerSoil Pro. PM: PowerMax Soil. * p < 0.05. ** p < 0.01.

t-test p = 0.041). PM also displayed 7-fold higher extraction yields in EstFS compared to PSP ($0.78 \pm 0.2$ µg$_{DNA}$ g$_{dw}^{-1}$, t-test p = 0.022). In EstCS, LM yielded $0.46 \pm 0.1$ µg$_{DNA}$ g$_{dw}^{-1}$, significantly more compared to PSP and PM (by 8-fold, t-test p = 0.01, and by 2.6-fold, t-test p = 0.037, respectively). Finally, at MANG, PSP yielded more DNA with $18 \pm 9$ µg$_{DNA}$ g$_{dw}^{-1}$, however not significantly different than LM and PM with $4.2 \pm 1$ and $5.3 \pm 0.3$ µg$_{DNA}$ g$_{dw}^{-1}$, respectively (t-test p > 0.05).

In terms of DNA concentrations and extraction yield, LM outperformed PSP and PM for HARB and the two sandy sediments studied.

### DNA integrity and size

The quality of DNA can be evaluated by DNA Integrity Number (DIN) or by its fragment size, both retrieved from automated electrophoresis analysis. DIN values typically range from 1 to 10 with increasing integrity. In EstFS, DIN values were higher with LM than with PM extraction (average of 9.1 versus 7.2, S1 Fig). DIN were lower when extracted using LM compared to PSP and PM in HARB (average of 3.5 versus 6.2 and 6.1, respectively). DIN values were similar in MANG between methods (averaging 5.6). DIN could not be retrieved for sediments EstCS and EstFS (extracted with PSP) due to insufficient DNA concentrations.

Fragment sizes, another proxy of DNA quality, varied between methods, ranging from 7,794 to > 60,000 bp (Fig 3). Fragment sizes were significantly higher in EstFS and EstCS when DNA was extracted using LM, compared to PM or PSP. Indeed, for EstFS, fragments extracted with LM reached a size of $57,500 \pm 2,000$ bp with one replicate above size limit of 60,000 bp. Those fragment sizes were significantly higher than PSP ($14,400 \pm 50$ bp, t-test p < 0.001) and PM ($27,500 \pm 3,600$ bp, t-test p < 0.001). For EstCS, fragment sizes of LM averaged at $55,600 \pm 5,000$ bp (with also one replicate above 60,000 bp), this value is significantly higher of those of PM ($19,500 \pm 1,500$ bp, t-test p = 0.003). PSP samples for EstCS did not contain enough DNA for analysis. For MANG sediments, PM extracted fragments of significantly higher sizes ($24,500 \pm 1,900$ bp) compared to LM and PSP ($14,300 \pm 820$ bp, t-test p = 0.005 and $8,750 \pm 650$ bp, p < 0.001 respectively). However, LM also extracted DNA of significant higher sizes than with PSP (t-test p = 0.016). For HARB, there was no significant differences in fragment sizes between methods, ranging from 10,300–21,000 bp.

Sandy sediments extracted with LM displayed up to 4-fold higher DNA fragment sizes compared to the other methods. For MANG, fragment sizes were decreasing with PM, LM and PSP respectively.

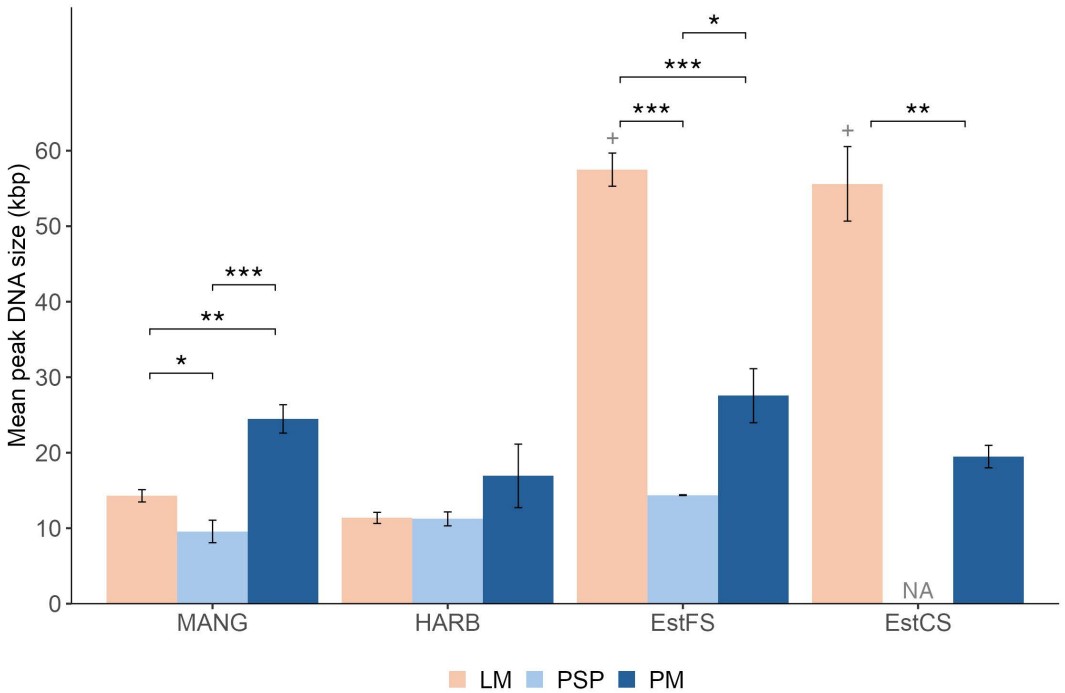

**Fig 3. Mean peak DNA size of LM, PSP and PM across sediments.** "+" indicates that one replicate was above size limit of 60,000 bp. "NA" indicates that the size analysis could not be performed due to insufficient DNA concentration. Statistical significance was assessed using a t-test. LM: Laboratory-made method. PSP: PowerSoil Pro. PM: PowerMax Soil. * p < 0.05. ** p < 0.01. *** p < 0.001.

## Amplification success and sequencing results

**PCR amplification success.** The success of PCR amplification for taxonomic markers varied across methods and sediments (Table 2). All 16S rRNA PCR amplification were successful for all sediments and methods. Almost all 18S rRNA PCR were successful, except for sediment EstCS, where 18S rRNA gene amplified only with LM and PM. However, with PM, the PCR did not yield enough product to allow for MiSeq sequencing.

## Alpha diversity

Alpha diversity metrics (i.e., ASV richness, Shannon Index and Simpson Inverse) revealed very few differences between methods for each sediment (S2 Fig). For prokaryotes, richness (ranging from 526 to 1,408 ASVs), Shannon Index (ranging from 5.7 to 6.6) and Simpson Inverse (ranging from 125 to 489) did not show any significant effect of extraction

**Table 2. PCR amplification results on 16S rRNA and 18S rRNA genes.**

| Sediment | MANG | | HARB | | EstFS | | EstCS | |
|---|---|---|---|---|---|---|---|---|
| *Target Gene* | 16S | 18S | 16S | 18S | 16S | 18S | 16S | 18S |
| **LM** | + | + | + | + | + | + | + | + |
| **PSP** | + | + | + | + | + | + | + | − |
| **PM** | + | + | + | + | + | + | + | +* |

LM: Laboratory-made method. PSP: PowerSoil Pro. PM: PowerMax Soil. + Positive amplification. - Negative amplification. * Positive PCR amplifications however Illumina MiSeq sequencing could not be performed due to insufficient PCR product.

method across sediments (ANOVA p > 0.05). However, differences in Simpson Inverse were sediment specific. Simpson Inverse was significantly higher with LM compared to PSP in MANG (t-test p = 0.01) and higher than both PSP and PM in EstCS (t-test p = 0. 046 and p = 0.046, respectively). However, Simpson Inverse was lower with LM compared to PSP in HARB (t-test p = 0.017).

For eukaryotes, richness ranged from 109 ASVs to 768 ASVs and significantly differ between methods (ANOVA p = 0.021). Post-hoc test revealed that PM displayed higher eukaryotic richness than LM across sediments (Tukey p = 0.023). In MANG, LM showed significantly lower richness (387 ± 39 ASVs) than PSP (553 ± 54 ASVs, t-test p = 0.018) and PM (729 ± 34 ASVs, t-test p = 0.001). For eukaryotes, Shannon Index and Simpson Inverse did not significantly differ between methods across sediments (ANOVA p > 0.05). However, differences were observed in MANG: Shannon Index and Simson Inverse were significantly lower with PSP compared to LM and PM (respectively t-test p = 0.009 and p = 0.005 for Shannon Index, p = 0.012 and p < 0.001 for Inverse Simpson).

There were no overall differences in alpha diversity of prokaryotes between methods across the studied sediments. For eukaryotes, the PM kit displayed higher overall richness than LM, yet no differences in evenness were found across the community. Most differences were sediment-specific, with instances where prokaryote and eukaryote evenness metrics were lower with PSP compared to LM (S2 Fig).

## Interindividual variability in community structure

Beta diversity measurements (i.e., phylogenetic distances) are very often used to compare groups of samples in microbial ecology. It allows quantification of differences in taxonomic structure of communities. The different DNA extraction methods used in this study resulted in differences of community structure. Indeed, for prokaryotes, DNA extraction methods influenced the structure of the community for each sediment (PERMANOVA on weighted UniFrac distances for MANG: p = 0.006; HARB, p = 0.006; EstFS, p = 0.016; EstCS, p = 0.005). For eukaryotes, extraction methods significantly influenced unweighted UniFrac distances between samples of MANG (PERMANOVA p = 0.004) and EstFS (PERMANOVA p = 0.011) but not HARB (PERMANOVA p = 0.217). Although the choice of extraction protocol influenced community structure within each sediment type, overall community composition remained clearly differentiated between sediment types regardless of the protocol used (see PCoAs in S3 Fig). Comparing phylogenetic distances between samples allow a quantitative assessment of interindividual variability in community structure. For a given sediment, median phylogenetic distances proved to be systematically higher between methods than within each method, confirming the influence of DNA extraction method on community structure (Fig 4). For prokaryotes, interindividual variability of each method was not statistically different across sediments (ANOVA p > 0.05), despite differences in MANG and HARB. Indeed, variability within PM was higher than variability within PSP and LM in MANG (t-test p = 0.041 and p = 0.01, respectively). In HARB, within-PSP variability was lower than variability within LM and PM (t-test p = 0.007 and p < 0.001, respectively). For eukaryotes, interindividual variability of each method was different across sediments (ANOVA p = 0.037). Post-hoc comparisons indicated that variability within PM was significantly lower than variability within LM (Tukey p = 0.029).

The analysis of phylogenetic distances proved that the extraction method influences community structure. For the prokaryotic communities, interindividual variability differences were sediment-specific whereas eukaryotic communities showed lower variability within PM compared to the LM method.

## Prokaryotes discriminantly enriched or depleted by LM

Beta diversity metrics indicated differences in taxa composition between methods. To evaluate those differences and give insights of which taxa were either enriched or lost by a method, a linear discriminant analysis effect size (LEfSe) was performed. It determines which ASVs are most likely responsible for differences between methods. For prokaryotes, LEfSe resulted in 103 ASVs and 109 ASVs respectively enriched and depleted in LM (compared to PSP and PM), representing 2.3% of total ASVs. Relative abundance of enriched ASVs in LM ranged from 0% to 0.62% (average of 0.009% across

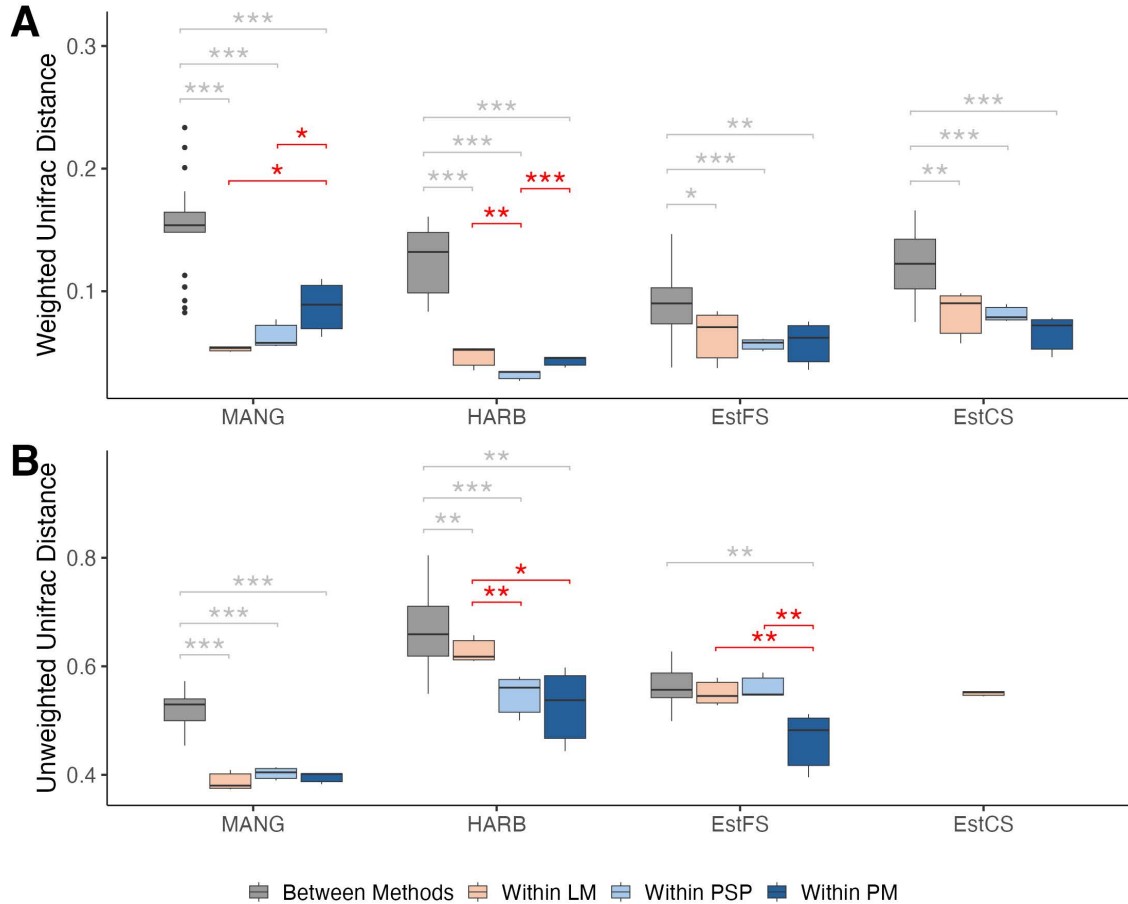

**Fig 4. Interindividual variability in community structure across sediments.** Phylogenetic distances between samples relates to interindividual variability in community composition. (A) Weighted UniFrac distances of prokaryotic taxa. (B) Unweighted UniFrac distances of eukaryotic taxa. Grey stars indicate significant differences between average within-method and between-method interindividual distances. Red stars indicate significant differences in average within-method interindividual distances across methods. Statistical significance was assessed using a t-test. LM: Laboratory-made method. PSP: PowerSoil Pro. PM: PowerMax Soil. * p<0.05. ** p<0.01. *** p<0.001.

sediments and methods). Depleted ASVs displayed higher relative abundance, ranging from 0% to 1.1% (average of 0.013%). The mean relative abundance of those depleted ASVs was also higher than the average relative abundance of all ASVs (average of 0.008%, ranging from 0% to 5.6%).

To summarise prokaryotes either enriched or reduced by LM compared to PSP or PM, a heat tree was generated to display median differences in relative abundance between LM and the other two methods for LEfSe-selected ASVs (Fig 5A). Across the community, LM selectively enriched ASVs from genera SEEP-SRB1, *Flavirhabdus*, *Altericroceibactrium*, IheB3-7, Sva0081 sediment group, Cm1–21, *Pirellula* and *Mariniblastus* (ordered by decreasing maximum LDA scores). Relative abundance of ASVs belonging to SEEP-SRB1 (family *Desulfosarcinaceae*) ranged from 0% to 0.6% across sediments. One ASV drove differences between methods, by being enriched in MANG with LM by 4.6-fold. One ASV affiliated to *Flavirhabdus* (family *Flavobacteriaceae*) was only detected in EstFS and EstCS with mean proportions of 0.24% and 0.09% respectively. It was enriched by LM in EstCS by a 2.6-fold increase in mean relative abundance. One ASV from genus *Altericroceibacterium* (family *Sphingomonadaceae*) was only detected in MANG and EstFS with proportions up to 0.21%. LM was the only method to detect this ASV. Four ASVs affiliated to IheB3-7 (family *Melioribacteraceae*)

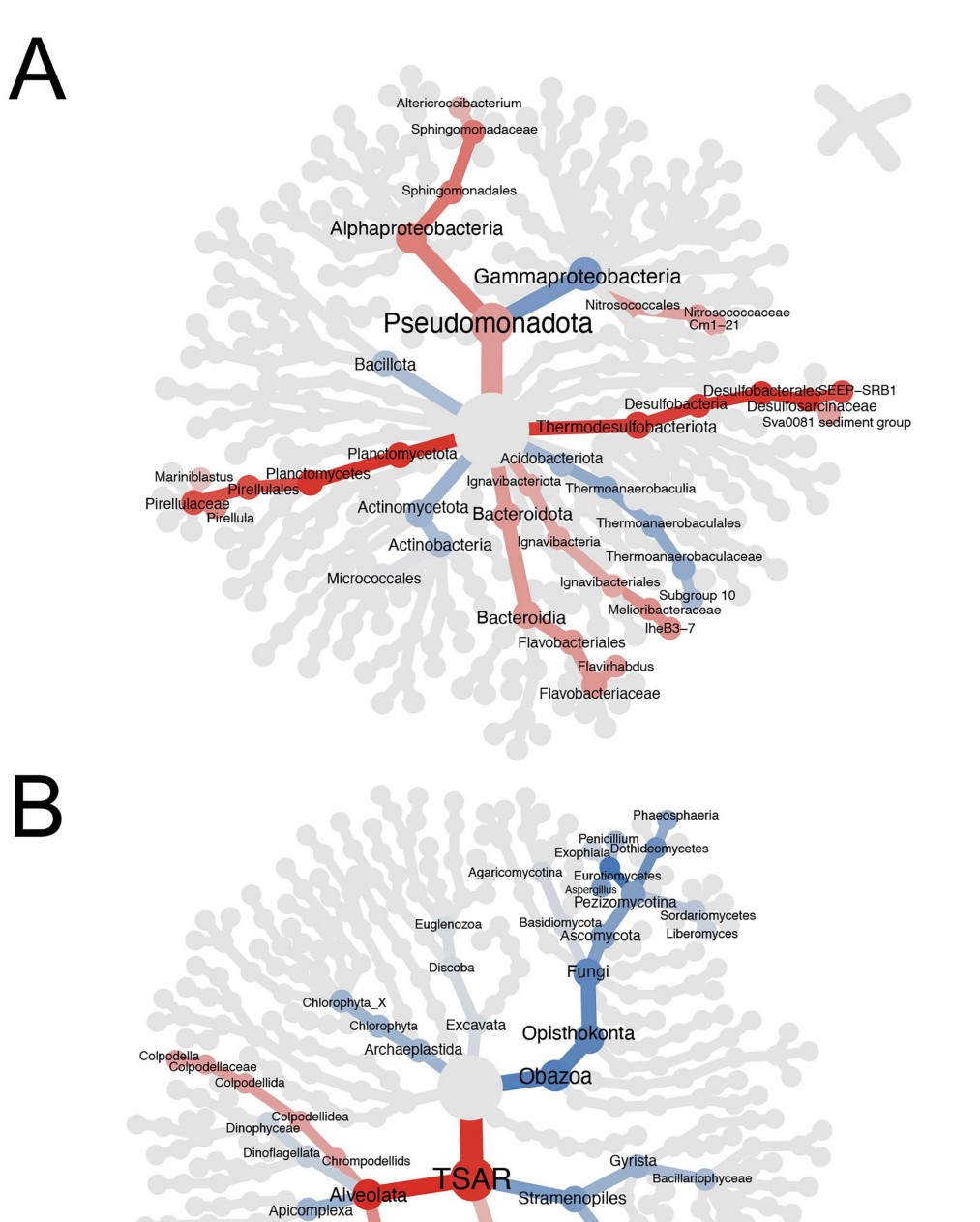

**Fig 5. Heat trees of enriched or depleted taxa by LM for prokaryotes and eukaryotes.** Median difference in proportion between Laboratory-made method (LM) and the other two methods (PowerSoil Pro and PowerMax Soil) of prokaryotic taxa (A) and eukaryotic taxa (B) at the genus level across all sediments. Selected ASVs by LEfSe (LDA>2, p<0.05) are either enriched (red) or depleted (blue) in LM. The eukaryotic data from sediment EstCS was unique to the LM method and therefore excluded from the LEfSe analysis.

displayed relative abundance ranging from 0 to 0.37%. They were enriched in EstFS and EstCS by LM by an average of 2.1-fold. ASVs affiliated with Sva0081 sediment group (family *Desulfosarcinaceae*) were only detected in MANG and HARB with relative abundance up to 0.28%. On the total 77 ASVs representing this genus, 3 were only detected in HARB using the LM method. One ASV affiliated with Cm1–21 (family *Nitrosococcaceae*) was found to be enriched in LM by an average of 1.5-fold. It was only found in EstFS and EstCS with relative abundance up to 0.18%. Four ASVs affiliated to *Pirellula* (family *Pirellulaceae*), that were only detected in EstFS and EstCS were enriched in LM by 2.4-fold and 3.5-fold, respectively, with relative abundance up to 0.17%. In the same family *Pirellulaceae*, ASVs affiliated with *Mariniblastus* were enriched in LM in EstFS by 3.6-fold (average relative abundance of 0.09%), where they were not detected with PSP and PM in MANG and EstCS.

Depleted prokaryotic ASVs by LM (compared to PSP and PM), exhibiting the highest differences in proportions between methods, were affiliated to different taxonomic levels. By descending order of maximum LDA, class *Gammaproteobacteria*, phylum *Bacillota*, order *Micrococcales*, and genus Subgroup 10 (belonging to family *Thermoanaerobaculaceae*) were reduced compared to Qiagen kits. Sixty-six ASVs affiliated with class *Gammaproteobacteria* (1,875 ASVs total) were depleted on average by 2-fold with LM. Fourteen ASVs of phylum *Bacillota* (out of 211 ASVs total) were not recorded in MANG or HARB with LM. Those ASVs were affiliated to families *Bacillaceae*, *Clostridiaceae*, *Lachnospiraceae*, *Oscillospiraceae*, *Peptostreptococcaceae*, and an unknown order of *Bacilli*. Four out of 33 ASVs affiliated with the order *Micrococcales* were not recorded in EstCS with LM. They are affiliated to families *Micrococcaceae*, *Microbacteriaceae* and *Beutenbergiaceae*. Nine out of 82 ASVs belonging to Subgroup 10 (family *Thermoanaerobaculaceae*) were depleted by 12-fold in EstFS with LM. They were not recorded in HARB with LM when relative abundance ranged from 0% to 0.08% with PSP (also not detected with PM).

To further illustrate shared and unique ASVs between methods, an Upset plot (S4A Fig) indicates that a large fraction of ASVs was shared across all three extraction methods. In addition to this shared fraction, each method recovered unique ASVs (318 for PSP, 285 for PM and 258 for LM). Notably, the LM method showed important overlap with the PM kit (with 264 shared ASVs).

Overall, approximately the same amount of prokaryotic ASVs were enriched and depleted in LM compared to the other methods. Enrichments were not sediment specific and were essentially affiliated to rare taxa. Depletions were mostly targeting more abundant taxa, with only a small proportion of ASVs for each taxon.

## Eukaryotes discriminantly enriched or depleted by LM

For eukaryotes, LEfSe resulted in 39 ASVs and 94 ASVs respectively enriched and depleted in LM compared to PSP and PM, representing 1.9% of total ASVs. Relative abundance of enriched ASVs in LM ranged from 0% to 0.65% (average of 0.016%). Depleted ASVs displayed higher relative abundance, ranging from 0% to 4.9% (average of 0.032%). ASVs that were either enriched or depleted with LM displayed higher average relative abundance than all ASVs (average of 0.012%, ranging from 0% to 14.8%).

A heat tree illustrating the median differences in eukaryotic proportions highlights taxa that are enriched and depleted with LM compared to the other methods (Fig 5B). Enriched taxa include ASVs affiliated with subdivision *Ciliophora*, genus *Colpodella* and class *Filosa-Thecofilosea* with maximum LDA scores of 3.13, 3.10, and 3.05 respectively. A total of 38 ASVs affiliated with *Ciliophora* (out of 614 ASVs in total) were enriched by 2.4-fold predominantly in EstFS with relative abundance up to 0.45%. Selected ASVs affiliated with *Colpodella* were enriched in HARB and EstFS by 2.3-fold and 4.9-fold respectively, with relative abundance up to 0.6%. ASVs affiliated with *Filosa-Thecofilosea* were predominant in HARB where LM enriched these ASVs by 3.5-fold, reaching relative abundances of 0.51%.

Depleted ASVs with median differences between LM and the other methods were affiliated to different taxonomic levels (Fig 5B). By decreasing maximum LDA score, those ASVs were affiliated to genus *Penicillium*, genus *Aspergillus*, class

*Dinophyceae*, subdivision *Apicomplexa*, genus *Phaeosphaeria*, subdivision *Euglenozoa*, an unknown subdivision of *Chlorophyta*, order *Agaricomycotina*, genus *Exophiala*, class *Bacillariophyceae*, an unknown family of *Chromadorea*, order *Labyrinthulomycetes*, family *Agaricomycetes* and genus *Liberomyces*. Most depleted ASVs were affiliated to fungi with a decrease in relative abundance of selected ASVs in MANG, HARB and EstFS by 7.7-fold, 9.5-fold and 2.5-fold respectively. However, those ASVs represent a fraction of fungal ASVs as LEfSe analysis selected 56 ASVs out of 697 in total. Similarly, other depleted ASVs represent only a fraction of their affiliated group, less than 7.3% of ASVs in their respective group. Their maximum relative abundance is less than 1%, except for *Dinophyceae* that were recorded up to 1.48% in HARB.

To further illustrate shared and unique ASVs between methods, an Upset plot (S4B Fig) indicates that, similarly to the prokaryotic community, a large fraction of ASVs was shared across all three extraction methods. In addition to this shared fraction, each method recovered unique ASVs (251 for PM, 220 for PSP and 171 for LM). The LM method showed important overlap with the PM kit (with 137 shared ASVs).

Eukaryotic taxa discriminantly enriched or depleted by LM compared to Qiagen kits were on average more abundant than all ASVs. They were less taxa enriched than depleted by LM compared to PSP and PM. Globally, enriched taxa were not sediment specific unlike depleted taxa where most differences were recorded in MANG and HARB. Depleted ASVs constituted a small fraction of their affiliated taxa.

### Cost analysis between methods

As of May 2025, catalogue price in France for Qiagen DNEasy™ PowerSoil™ Pro is given at €561, enough to process 50 samples. Qiagen DNEasy™ PowerMax™ Soil is far more expensive per sample, with €494 for 10 samples. The LM method on the other hand, have a higher starting cost of €1,622 but is sufficient to process up to 1,000 samples (based on reagent and tubes costs, detailed in S2 File). This makes the LM method the least expensive per sample compared to the two commercial kits (Table 3). Even if the LM method need twice the labor time, these cost differences may be worth considering.

## Discussion

### Long DNA fragments from sandy sediments best extracted with the LM method

Our results show that DNA extraction yields and fragment sizes vary between muddy and sandy sediments. For muddy sediments, no general trend between methods indicates that the efficiency of methods was sediment specific. For sandy sediments, the LM method outperforms both commercial kits in terms of DNA extraction yield up to a factor of 13. Those results demonstrate the high efficiency of cell lysis, commonly better with enzymatic lysis compared to mechanical disruption of cells [15,16,38]. DNA integrity and size, was also found to be much higher with LM in sandy sediments (up to 4 times the fragment size), highlighting that the method yields high-quality DNA. This is consistent with previous findings that bead-beating and purification columns of commercial kits reduce molecular weight [16,39]. Method LM was also the

**Table 3. Costs for each method based on reagents and tubes as of May 2025.**

| Method | LM | PSP | PM |
|---|---|---|---|
| Starting cost (€) | €1,622.00 | €561.00 | €494.00 |
| Number of samples | 1,000 | 50 | 10 |
| Price per sample (€) | €1.62 | €11.22 | €49.40 |
| Labor time (h) | 8 (+2)* | 3 | 4 |

LM: Laboratory-made method. PSP: PowerSoil Pro. PM: PowerMax Soil. * LM needs approximately 2h of preparation of reagents and buffer.

only method to successfully amplify 18S rRNA gene with sediment EstCS and yield enough PCR product for NGS. The PSP and PM kits yielded DNA concentrations below 1 ng µL$^{-1}$, likely insufficient for successful PCR amplification of the studied taxonomic marker for eukaryotes.

The LM method did not display differences in prokaryotic alpha diversity metrics compared to commercial kits. Differences in eukaryotic ASV richness were recorded across sediments, but it did not affect relative abundance or distribution of taxa as evenness metrics were consistent across methods. This indicates that the LM method is the best method for extracting DNA from sandy sediments. With higher DNA fragment sizes and high extraction yields, those are first steps towards long-read sequencing, that could be implemented in sandy sediments with higher starting masses. Long-read sequencing can facilitate taxonomic affiliations and the exploration of gene repertoires through metagenomics, contributing to a better understanding of ecosystem functioning and biotechnological potential of samples that are difficult to process [9].

## Taxonomic biases induced by extraction methods

It is well documented that DNA extraction method affects community composition [40,41]. With similar findings in our study, it emphasizes the need for consistency in extraction methods when comparing studies. Despite differences in extraction methods, beta diversity analysis showed that sediments were clustered by site regardless of the extraction method, with most taxa being shared across methods. Nonetheless, each method recovered a substantial number of unique ASVs in both prokaryotic and eukaryotic datasets, likely reflecting differences in lysis chemistry and input sediment masses. Indeed, it was previously found that starting masses affect community composition [41–43]. The PM kit uses 50 times more starting material than LM, most likely explaining the observed differences in eukaryotic communities. Indeed, the PM kit displayed overall higher eukaryotic richness and lower interindividual variability compared to the LM method. This hypothesis is reinforced by the depleted richness and abundance in fungal taxa with the LM method, as it was shown that 10 g of starting material inferred higher fungal richness compared to lower masses [41]. However, with 250 mg of starting material with PSP, more fungal taxa were detected compared to PM (414 ASVs versus 279 ASVs), indicating that starting mass is not the only factor. Moreover, Penton et al. (2016) [41] also found that lower amounts of starting material induced higher interindividual variability, this was not the case in our study. Indeed, reproducible profiles were recorded for prokaryotic communities, where differences were rather sediment-specific than method-induced. For eukaryotic communities, the lack of overall differences between LM and PSP highlights the fact that interindividual variability seems more regulated by starting masses than method. This suggests that the LM method is reproducible and can be used with small quantities of starting material, which is particularly useful for specific sampling strategies that yield limited sample volumes.

From the LEfSe analysis, the commercial kits seem to selectively enrich Gram-positive bacteria compared to the LM method. This method-induced difference may be due to the difficulty of biochemical lysis to disrupt the peptidoglycan layer of Gram-positive bacteria, or, conversely, the ability of powerful mechanical lysis to break bacterial spores [44]. To improve DNA extraction in Gram-positive bacteria, the lysis steps of LM can be extended. However, the selective enrichment by commercial kits may be linked not only to lysis differences, but also to reagent DNA contamination originating from the kits, which can affect observed microbial communities [45].

Despite those differences, the LM method enriched rarer taxa than commercial kits. This indicates the possibility of studying the rare biosphere with this method, often overlooked despite its importance in ecosystems [46,47].

## Implications and recommendations

As demonstrated by this work, DNA extraction method can influence microbial community composition, particularly regarding detection of rare taxa. For studies focused on comparative microbial ecology, LM presents a robust alternative to preexisting extraction methods with excellent yields, high fragment integrity, reproducible profiles and high selection

of rare taxa. The variability observed across sediments converges with previous work to demonstrate that no extraction method is optimal for all sediment types [17] and that the method should ideally be validated for the sample type of interest. The LM method is particularly recommended for sandy and putative low-biomass sediments, and for studies requiring high-integrity DNA for long-read sequencing. To meet requirements for long-read sequencing (e.g., DNA concentration and mass), the LM method can easily be adapted for a higher amount of starting material. For potentially high-biomass sediments, with equivalent or better yield and fragment sizes than PSP, the LM method can be a cost-effective alternative to commercial kits, with little to no impact on diversity metrics. This can be useful for developing countries or long-term series that require a large number of samples.

## Conclusion

DNA extraction method substantially influences both the quantity and quality of DNA retrieved from coastal sediments, as well as the microbial community composition studied by high-throughput sequencing. The proposed the LM method yielded best results in sandy environments, notably by extracting large amounts of high-molecular-weight DNA. Very few differences in community composition make this method a cost-effective alternative to pre-existing ones.

## Supporting information

**S1 File. Details on optimization of the LM method.**
(DOCX)

**S2 File. Reagent and tubes costs of the LM method.** * Catalogue prices in France as of May 2025. CTAB: hexadecyl-trimethylammonium bromide. EDTA: ethylenediaminetetraacetic acid.
(XLSX)

**S1 Fig. Genomic DNA ScreenTape results for the different sediments.** (A) Laboratory-made method (LM). (B) Power-Soil Pro (PSP). (C) PowerMax Soil (PM).
(PDF)

**S2 Fig. Alpha diversity metrics comparing LM, PSP and PM across different sediments.** Diversity metrics include ASV richness, Shannon Index, and Inverse Simpson Index (InvSimpson). The left panel represents prokaryotic communities, while the right panel represents eukaryotic communities. Statistical significance was assessed using a t-test. LM: Laboratory-made method. PSP: PowerSoil Pro. PM: PowerMax Soil. * $p < 0.05$. ** $p < 0.01$. *** $p < 0.001$.
(PDF)

**S3 Fig. PCoAs of microbial communities across sediments and extraction methods.** (A) PCoA on weighted UniFrac distances of prokaryotic taxa. (B) PCoA on unweighted UniFrac distances of eukaryotic taxa. LM: Laboratory-made method. PSP: PowerSoil Pro. PM: PowerMax Soil.
(TIFF)

**S4 Fig. Upset plot displaying shared and unique ASVs between extraction methods.** (A) Prokaryotic ASVs. (B) Eukaryotic ASVs. Shared and unique ASVs were defined based on presence in all three replicates of a given sediment for each method.
(TIFF)

**S1 raw images. Original uncropped gel images presented in S1 File.**
(TIF)

## Acknowledgments

We would like to thank Sophie Guasco and Léa Sylvi for their help with automated electrophoresis analyses.

## Author contributions

**Conceptualization:** Benjamin Misson, Elisabeth Navarro.

**Investigation:** Leopold Matthys, Eléonord Mayissah Moungues, Anaëlle Genève, Elisabeth Navarro.

**Visualization:** Leopold Matthys.

**Writing – original draft:** Leopold Matthys.

**Writing – review & editing:** Leopold Matthys, Eléonord Mayissah Moungues, Anaëlle Genève, Virginie Sanial, Benjamin Misson, Elisabeth Navarro.

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
