## [Decision Letter · Decision Letter 0]

21 Nov 2025

Dear Dr. Navarro,

Thank you for submitting your manuscript to PLOS ONE. After careful consideration, we feel that it has merit but does not fully meet PLOS ONE’s publication criteria as it currently stands. Therefore, we invite you to submit a revised version of the manuscript that addresses the points raised during the review process.

We look forward to receiving your revised manuscript.

Kind regards,

Raffaella Casotti

Academic Editor

PLOS ONE

Journal Requirements:

Additional Editor Comments:

Dear Authors, I regret to inform you that the manuscript cannot be accepted as such. However, the three referees appreciated the effort and the approach adopted. Therefore, it will be possible for you to revise or rebutt the criticisms raised in a modified version of the manuscript. Please, consider that the manuscript will undergo another round of revision.

Thank you for submitting your best work to our Journal.

Reviewers' comments:

Reviewer's Responses to Questions

**Comments to the Author**

1. Is the manuscript technically sound, and do the data support the conclusions?

Reviewer #1: Yes

Reviewer #2: Partly

Reviewer #3: Yes

2. Has the statistical analysis been performed appropriately and rigorously?

Reviewer #1: Yes

Reviewer #2: I Don't Know

Reviewer #3: Yes

3. Have the authors made all data underlying the findings in their manuscript fully available?

Reviewer #1: Yes

Reviewer #2: Yes

Reviewer #3: Yes

4. Is the manuscript presented in an intelligible fashion and written in standard English?

Reviewer #1: Yes

Reviewer #2: Yes

Reviewer #3: Yes

Reviewer #1: The authors present a laboratory-made (LM) DNA extraction method benchmarked against two commercial kits for coastal sediments. The LM approach produced substantially higher yields and longer DNA fragments, enabling improved recovery of eukaryotic diversity. Although more labor-intensive, it proved reproducible, cost-effective, and well-suited for large-scale or resource-limited studies. Overall, the paper makes a valuable contribution, but certain sections (methods clarification, conceptual definitions, and expanded comparisons) should be improved for clarity and completeness.

Abstract

The sentence “While prokaryotic diversity metrics were largely similar across methods, we observed differences in eukaryotic richness that were likely influenced by the amount of sediment processed” could be worth keeping, but you should clarify why this observation is relevant for the paper’s scope. Does it support the strength of your method (better for eukaryotes)? Or does it show a limitation (sensitive to sample size)? At present, it feels underdeveloped.

Introduction

Line 49: The sentence “However, those technologies are still dependent on DNA extraction methods” — but isn’t this true for other sequencing or analysis approaches as well? Please clarify the distinction you are making here, otherwise the statement sounds too general.

Line 55: Since this is a generalist journal, please briefly define alpha and beta diversity for non-specialist readers (one short clause each should be enough).

Please expand slightly on the difference between NGS shotgun sequencing and metabarcoding (at least two sentences). For example: emphasize that metabarcoding targets specific loci (e.g., 16S, COI), which can bias taxonomic resolution, whereas shotgun metagenomics enables recovery of longer fragments and broader taxonomic/functional profiles.

Methods

Line 61: Please describe the sediment type in more detail (grain size, organic content, coastal location, depth collected, etc.). This information is important to contextualize yield and inhibition challenges.

Line 77: Clarify what exactly was performed in triplicate (DNA extractions? PCRs? sequencing libraries?). Also, explain why triplicates were necessary (e.g., to ensure reproducibility, to capture variability within sediment aliquots).

Line 93: The statement “protocol should be adapted” is vague. If you mean that increasing input material often leads to inhibitor co-extraction, please state that explicitly and explain what adaptations you recommend or tested.

Line 175: “Clenned?” — please check this typo. If you mean cleaned libraries, specify the parameters: bead-to-sample ratios, fragment size ranges retained, or kits used. This will improve reproducibility.

Line 200: The authors already mention “Kruskal–Wallis p-value cutoff of 0.05” (line 195). Later (line 200), you repeat “p-value threshold for statistical significance was 0.05”. Consider consolidating into one clear statement at the end.

Results

Line 211: When mentioning statistically significant or nt, include p-val and test.

Discussion

Consider adding a brief statement about limitations (e.g., only sandy sediments tested, other sediment types may behave differently).

Reviewer #2: The paper « Cost-effective DNA extraction method optimized for high yield and long fragments from

coastal sediments » proposes a new « home-made » DNA extraction method efficient for sediments, particarly for sandy or low-biomass sediments. Moreover, this method could be very useful for studies requiring high-quality DNA, specifically for long-fragments analysis, like metabarcoding or metagenomics using third generation sequencing technologies.

This protocol was compared to very widespread commercial kits used in genomic environmental studies, and different soil compositions were tested, which was very important for the protocol validation. Moreover, DNA extractions were realized in triplicate for each method to check the repetability and reproductibility of the lab work.

To check the efficiency of the method, 3 parameters have been checked : DNA extraction yield, DNA fragment size, and alpha and beta biodiversity based on 16S (prokaryotes) and 18S (eukaryotes) markers.

I really appreciated the details in the protocol, which can be directly applied in the lab. The references and the costs are clearly listed.

Moreover the methods used to measure the quantities and the qualities of DNA extracts are really precises and showed in the paper.

Nevertheless, I found the protocol quite time-consuming and requires long reagent preparation time steps. These 2 factors have to be considered when you have a lot a samples to process.

I also have some questions an remarks about the strategy of study validation :

1- Did you try to modify some parameters from commercial kits PSP and PM ? I know lysis steps can be very rough. Did you test different lysis condition sto to optimize the integrity of DNA extracts?

2- I am quite sceptical regarding eukaryotes PCR amplification step :

- Go Taq polymerase (Promega) is known to not be the best choice to amplify marine microbes, specifically protists. Most of protists taxa have high-rich GC content, so Kapa and Phusion polymerases are preferably used.

- Could you explain the choie of V7 marker and primers please ? I totaly approve V7 is a good candidate regarding sedaDNA metabarcoding studies, but n most of modern marine eukaryotic metabarcoding studies (coastal and open ocean areas), V4 or V9 markers are more used. (ie Zimmermann et al. 2024 DOI:10.1002/edn3.580, Mahe et al. 2017 DOI:10.1038/s41559-017-0091).

3- How many replicates did you do for each of your DNA amplifications ? I didn’t find the information in the text.

4- You highlight the potential of the LM method to recover long-fragments of DNA. Why didn’t you check alpha and beta diversity using 3rd generation sequencing like PacBio or ONT ? (ie https://doi.org/10.1101/2025.07.20.665787,
https://doi.org/10.3897/mbmg.9.163750 )

To conclude, I would suggest to do more amplification tests on V7 marker and longer fragments (like entire 18S marker or more) and using a different Taq such as Phusion High Fidelity F532L to prove the efficiency of the method.

Reviewer #3: This manuscript presents a laboratory-made (LM) DNA extraction method optimized for high yield and long DNA fragments from coastal sediments, comparing its performance against two commercial Qiagen kits (PowerSoil Pro and PowerMax Soil). The study addresses a relevant methodological bottleneck in environmental DNA research, extracting high-quality DNA from complex, low-biomass matrices such as sandy or organic-rich sediments, with the goal of providing an affordable, non-hazardous alternative suitable for large-scale or resource-limited settings.

The topic is scientifically relevant and of clear practical value. The study is generally well structured and clearly written. However, some aspects of the methodological comparability, data normalization, and interpretative depth limit the robustness of the conclusions. While the LM method appears promising, certain parts of the work require clarification, additional controls, and normalization to ensure fair comparisons among extraction approaches.

1- How many independent replicates were performed for each extraction protocol and sediment type? Please specify whether biological and technical replicates were included. This is essential to assess reproducibility and to validate the statistical comparisons (ANOVA, PERMANOVA).

2- No extraction blanks or negative controls appear to have been sequenced. These are essential for detecting contamination, especially in low-biomass matrices.

3- The authors appropriately normalized DNA yields to both sediment dry weight (µg DNA g⁻¹) and total elution volume, as stated in the Methods (lines 220–222). This normalization ensures that reported extraction efficiencies are mathematically comparable across the three methods (LM, PSP, and PM). However, even though yield normalization was properly performed, the biological and ecological comparability among extraction protocols remains partially limited due to the large differences in input sediment mass and lysis chemistry: LM and PSP processed 200–250 mg of sediment, whereas PM used 5 g, potentially capturing a broader microbial and eukaryotic diversity simply by integrating more microhabitats. The LM protocol relies on enzymatic and detergent-based lysis (CTAB/SDS + Proteinase K), while PSP and PM use mechanical bead-beating, which can favor the extraction of different cell types (e.g., Gram-positive bacteria, spores). These factors could influence both DNA yield and community composition, independent of extraction efficiency. Therefore, the manuscript should explicitly acknowledge in the Discussion that, while yield data were normalized correctly, the ecological representativeness of each extraction remains partly influenced by input mass and lysis mechanism. To strengthen this comparison, I recommend including a PCoA or NMDS plot showing beta-diversity differences between extraction protocols across sediment types. This would visually confirm whether extraction method drives distinct community structures.

4- It is unclear whether sequencing depth normalization or rarefaction was performed before comparing alpha and beta diversity. This is critical for metabarcoding data to ensure comparability across methods and sediments, The Methods should specify how read counts were normalized before ANOVA/PERMANOVA tests (e.g., rarefaction to minimum library size, relative abundance transformation, or DESeq2 variance stabilization). To complement this, an Upset plot illustrating shared vs unique ASVs among the different extraction protocols would provide a more intuitive visualization of overlap and method-specific recovery of taxa. This approach would highlight whether the LM method captures a unique subset of taxa or primarily overlaps with commercial kits.

**Do you want your identity to be public for this peer review?** For information about this choice, including consent withdrawal, please see our Privacy Policy

Reviewer #1: No

Reviewer #2: No

Reviewer #3: No

---

## [Author Response · Author response to Decision Letter 1]

2 Jan 2026

Responses to comments and questions of Reviewer 1:

1. The authors present a laboratory-made (LM) DNA extraction method benchmarked against two commercial kits for coastal sediments. The LM approach produced substantially higher yields and longer DNA fragments, enabling improved recovery of eukaryotic diversity. Although more labor-intensive, it proved reproducible, cost-effective, and well-suited for large-scale or resource-limited studies. Overall, the paper makes a valuable contribution, but certain sections (methods clarification, conceptual definitions, and expanded comparisons) should be improved for clarity and completeness.

Abstract

The sentence “While prokaryotic diversity metrics were largely similar across methods, we observed differences in eukaryotic richness that were likely influenced by the amount of sediment processed” could be worth keeping, but you should clarify why this observation is relevant for the paper’s scope. Does it support the strength of your method (better for eukaryotes)? Or does it show a limitation (sensitive to sample size)? At present, it feels underdeveloped.

Author response: Thank you for your comment, we meant it as a limitation of the method and the importance of sample size for the study of microeukaryotes. We thus changed the sentence for:

“While prokaryotic diversity metrics were largely similar across methods, we observed differences in eukaryotic richness that were likely influenced by the amount of sediment processed. This demonstrates the higher importance of sample amount when assessing microeukaryotic diversity.” Line 35.

2. Line 49: The sentence “However, those technologies are still dependent on DNA extraction methods” — but isn’t this true for other sequencing or analysis approaches as well? Please clarify the distinction you are making here, otherwise the statement sounds too general.

Author response: Thank you for your suggestion. We agree that the statement is indeed too general. The goal was to indicate the importance of DNA extraction as it is the first step before studying taxonomy using molecular tools. To clarify the sentence, we improved it as such:

“However, those technologies are still dependent on DNA extraction methods, indicating the importance of the efficiency of DNA extractions to recover high yields of high-quality DNA (long fragments and inhibitor free) and capture broader microbial diversity.” Line 62.

3. Line 55: Since this is a generalist journal, please briefly define alpha and beta diversity for non-specialist readers (one short clause each should be enough).

Author response: Thank you for your comment. We added a definition of alpha diversity “within-sample taxonomic richness and evenness” and beta diversity “between-sample compositional dissimilarity”. Line 71.

4. Please expand slightly on the difference between NGS shotgun sequencing and metabarcoding (at least two sentences). For example: emphasize that metabarcoding targets specific loci (e.g., 16S, COI), which can bias taxonomic resolution, whereas shotgun metagenomics enables recovery of longer fragments and broader taxonomic/functional profiles.

Author response: Thank you for your comment. We have expanded the description of the differences between metabarcoding and shotgun metagenomics in the revised manuscript:

“Metabarcoding targets specific genetic loci (e.g., fragments of 16S or 18S rRNA genes) to generate taxonomic profiles from established marker genes with high coverage of the genetic diversity for the targeted loci. In contrast, shotgun metagenomics sequences all DNA present in a sample, providing access to wider range of taxonomic and functional information, while providing less sequencing depth on each sequenced genomic region. Third-generation sequencing further improves shotgun metagenomics by producing long, continuous reads that help resolve complex regions and strengthen genome reconstruction. It also enables long-read metabarcoding, allowing full-length marker sequencing and improved taxonomic resolution.” Line 55-61

5. Line 61: Please describe the sediment type in more detail (grain size, organic content, coastal location, depth collected, etc.). This information is important to contextualize yield and inhibition challenges.

Author response: Thank you for this suggestion. To improve clarity for readers, we added the note “characteristics, location, organic and water content are described in Table 1” (Line 78). We believe the manuscript already includes key contextual information, namely:

(1) Coastal location: “in a coastal area (near Alembétogo, Gabon)” Line 82; “a small marina (Oursinieres harbor, Le Pradet, France)” Line 84; “a subterranean estuary located on Pellegrin Beach (La Londe-les-Maures, France)” Line 85.

(2) Sampling depth: “depth of 5 centimeters.” Line 82; “~ 0.2-meter depth” Line 87.

We trust directing readers to Table 1 for sediment characteristics, organic content, and water content provides the additional details you requested.

6. Line 77: Clarify what exactly was performed in triplicate (DNA extractions? PCRs? sequencing libraries?). Also, explain why triplicates were necessary (e.g., to ensure reproducibility, to capture variability within sediment aliquots).

Author response: Thank you for pointing this out. We added the sentence: “At each site, sediments were collected in three independent 50 mL Corning tubes to capture within-site variability.” (Line 89) and modified the sentence explaining how DNA extractions were performed “For each DNA extraction method, three separate DNA extractions were performed on three thawed environmental triplicates for each sediment.” Line 99.

7. Line 93: The statement “protocol should be adapted” is vague. If you mean that increasing input material often leads to inhibitor co-extraction, please state that explicitly and explain what adaptations you recommend or tested.

Author response: Thank you for your comment. We agree with the fact that increasing the starting mass increases the co-extraction of inhibitors. Therefore, we clarified the sentence: “To recover more DNA, LM can be adapted for larger amounts of starting material. To prevent increased co-extraction of inhibitors, lysis and precipitation volumes must be increased proportionally to maintain the same sediment-to-reagent ratio. For example, successful extractions were performed using 2 g of sediment in 6.6 mL of lysis buffer and CTAB in 15 mL Corning tubes.” Line 117.

8. Line 175: “Clenned?” — please check this typo. If you mean cleaned libraries, specify the parameters: bead-to-sample ratios, fragment size ranges retained, or kits used. This will improve reproducibility.

Author response: Thank you for your comment. Eurofins Genomics provided us – in their terminology – “cleaned” reads that were pre-processed using Cutadapt software. To clarify, we modified the two sentences as such: “Eurofins Genomics provided MiSeq reads that were trimmed of Illumina adaptors using Cutadapt software v2.7 [23]. Reads of 16S and 18S fragments were then separately processed […]” Line 211.

9. Line 200: The authors already mention “Kruskal–Wallis p-value cutoff of 0.05” (line 195). Later (line 200), you repeat “p-value threshold for statistical significance was 0.05”. Consider consolidating into one clear statement at the end.

Author response: Thank you for pointing this out. The first occurrence of this sentence was to describe the parameters used to perform the LEfSe analysis, and the second occurrence to generalize to other statistical tests. To improve intelligibility, we modified the first occurrence: “LEfSe parameters were the following: norm = “CPM”, taxa_rank = “none”, kw_cutoff = 0.05, lda_cutoff = 2.” Line 236.

10. Line 211: When mentioning statistically significant or nt, include p-val and test.

Author response: Thank you for pointing this out. We added the test and the p-value when we mentioned “significant” and either information was missing.

11. Consider adding a brief statement about limitations (e.g., only sandy sediments tested, other sediment types may behave differently).

Author response: Thank you for your comment. We added a brief statement in the Discussion to highlight the fact that results vary between muddy and sandy sediments we tested:

“Our results show that DNA extraction yields and fragment sizes vary between muddy and sandy sediments. For muddy sediments, no general trend between methods indicates that the efficiency of methods was sediment specific. For sandy sediments, the LM method outperforms both commercial kits in terms of DNA extraction yield up to a factor of 13.” Line 519.

Responses to comments and questions of Reviewer 2:

The paper « Cost-effective DNA extraction method optimized for high yield and long fragments from

coastal sediments » proposes a new « home-made » DNA extraction method efficient for sediments, particarly for sandy or low-biomass sediments. Moreover, this method could be very useful for studies requiring high-quality DNA, specifically for long-fragments analysis, like metabarcoding or metagenomics using third generation sequencing technologies.

This protocol was compared to very widespread commercial kits used in genomic environmental studies, and different soil compositions were tested, which was very important for the protocol validation. Moreover, DNA extractions were realized in triplicate for each method to check the repetability and reproductibility of the lab work.

To check the efficiency of the method, 3 parameters have been checked : DNA extraction yield, DNA fragment size, and alpha and beta biodiversity based on 16S (prokaryotes) and 18S (eukaryotes) markers.

I really appreciated the details in the protocol, which can be directly applied in the lab. The references and the costs are clearly listed.

Moreover the methods used to measure the quantities and the qualities of DNA extracts are really precises and showed in the paper.

Nevertheless, I found the protocol quite time-consuming and requires long reagent preparation time steps. These 2 factors have to be considered when you have a lot a samples to process.

I also have some questions an remarks about the strategy of study validation :

1. Did you try to modify some parameters from commercial kits PSP and PM ? I know lysis steps can be very rough. Did you test different lysis condition sto to optimize the integrity of DNA extracts?

Author response: Thank you for your comment. We did not modify lysis steps of the commercial PSP and PM kits. Reducing mechanical lysis would likely decrease DNA yield rather than improve fragment size (e.g., Ma et al. 2020, https://doi.org/10.3389/fmicb.2020.581227). Although such optimization could be explored in future work, our objective here was to develop a cost-effective and efficient alternative to commercial kits, particularly one suitable for use in developing countries.

2. I am quite sceptical regarding eukaryotes PCR amplification step :

2.a. Go Taq polymerase (Promega) is known to not be the best choice to amplify marine microbes, specifically protists. Most of protists taxa have high-rich GC content, so Kapa and Phusion polymerases are preferably used.

5. To conclude, I would suggest to do more amplification tests on V7 marker and longer fragments (like entire 18S marker or more) and using a different Taq such as Phusion High Fidelity F532L to prove the efficiency of the method.

Author response: Thank you for this constructive suggestion. We agree that polymerase choice can influence amplification success. Following your recommendation, we tested an alternative high-fidelity enzyme (Phusion High Fidelity Master Mix with GC Buffer, F532) with the same starting DNA amount and adjusted denaturation temperatures according to the manufacturer’s guidelines. However, this polymerase did not improve amplification success: samples that failed with GoTaq also failed with Phusion (Sediment PelCS extracted with PSP), and overall amplification rates across all samples were lower with Phusion than with GoTaq (19 vs. 33 positive amplifications out of 36 samples, respectively). These results suggest that amplification failures were not primarily driven by polymerase choice, but likely by sample-specific factors such as DNA integrity or inhibitor carry-over.

To document this comparison, we provide in the file named Response to Reviewers.docx the gel image showing amplification results obtained with the Phusion polymerase. For this electrophoresis, 5 µL of PCR product were loaded per well; migration was performed at 135V for 20min, using the FastGene 100bp DNA Marker (Nippon Genetics).

2.b. Could you explain the choie of V7 marker and primers please ? I totaly approve V7 is a good candidate regarding sedaDNA metabarcoding studies, but n most of modern marine eukaryotic metabarcoding studies (coastal and open ocean areas), V4 or V9 markers are more used. (ie Zimmermann et al. 2024 DOI:10.1002/edn3.580, Mahe et al. 2017 DOI:10.1038/s41559-017-0091).

Author response: Thank you for that interesting comment. As noted in Zimmermann et al. (2024), the V4 region generally provides higher taxonomic resolution than the V7 region. However, the V7 region remains a suitable marker in terms of broad taxonomic coverage – often outperforming the V9 region – and continues to be used in recent marine studies (e.g., https://doi.org/10.1016/j.funeco.2025.101460). Using shorter markers such as V7 or V9 can also help mitigate amplification challenges, particularly in low-biomass environments where longer fragments (e.g., V4) amplification may fail more frequently. Importantly, Zimmermann et al. (2024) was not yet available at the time we designed our study, but we plan to consider the V4 region in future work. While the choice of 18S markers can be discussed at length, we do not expect that using the V4 region in this study would have altered the conclusions regarding the performance of the proposed method.

3. How many replicates did you do for each of your DNA amplifications ? I didn’t find the information in the text.

Author response: Thank you for your question. Biological replicates were extracted separately to ensure reproducibility. Each DNA sample was then amplified once (i.e., no technical replicates was performed for the PCR amplification). To clarify the Methods section, we added information about PCR replicates in the “PCR amplification and sequencing” section: “Each DNA sample was amplified once; no technical PCR replicates were performed.” Line 199. We also clarified in the “Sediment collection” which procedures were performed in triplicates (Lines 89-91 and 99).

4. You highlight the potential of the LM method to recover long-fragments of DNA. Why didn’t you check alpha and beta diversity using 3rd generation sequencing like PacBio or ONT ? (ie https://doi.org/10.1101/2025.07.20.665787,
https://doi.org/10.3897/mbmg.9.163750 )

Author response: Thank you for your insightful comment. Indeed, third-generation sequencing technologies such as PacBio or ONT offer higher taxonomic resolution, which would enable more accurate comparisons of method-specific taxa. However, for basic diversity metrics, Bludeau et al. (2024) showed that metabarcoding using shorter markers than V7 (V9 in their case) yields beta-diversity patterns comparable to those obtained from full-length 18S rRNA sequencing with ONT (https://doi.org/10.1002/edn3.70084). Additionally, the implementation of long-read sequencing was not feasible within the resource constraints of this project.

Responses to comments and questions of Reviewer 3:

This manuscript presents a laboratory-made (LM) DNA extraction method optimized for high yield and long DNA fragments from coastal sediments, comparing its performance against two commercial Qiagen kits (PowerSoil Pro and PowerMax Soil). The study addresses a relevant methodological bottleneck in environmental DNA research, extracting high-quality DNA from complex, low-biomass matrices such as sandy or organic-rich sediments, with the goal of providing an affordable, non-hazardous alternative suitable for large-scale or resource-limited settings.

The topic is scientifically relevant and of clear practical value. The study is generally well structured and clearly written. However, some aspects of the methodo

---

## [Decision Letter · Decision Letter 1]

1 Feb 2026

Dear Dr. Navarro,

Thank you for submitting your manuscript to PLOS ONE. After careful consideration, we feel that it has merit but does not fully meet PLOS ONE’s publication criteria as it currently stands. Therefore, we invite you to submit a revised version of the manuscript that addresses the points raised during the review process.

**Please, see the reviewer's comments**

We look forward to receiving your revised manuscript.

Kind regards,

Raffaella Casotti

Academic Editor

PLOS One

Journal Requirements:

Additional Editor Comments:

Daer Authors, while the referee acknowledges that the majority of criticisms have been addressed, there still are some points that need to be clarified in the manuscript. And some of them require to smooth the statements of the overall conclusions, in terms of readiness and validity of the protocol proposed.

As a consequence, we cannot accept the revised version per se, but another round of corrections is needed.

Thank you for submitting your best work to PLoS

Reviewers' comments:

Reviewer's Responses to Questions

**Comments to the Author**

Reviewer #2: All comments have been addressed

2. Is the manuscript technically sound, and do the data support the conclusions?

Reviewer #2: Yes

3. Has the statistical analysis been performed appropriately and rigorously?

Reviewer #2: Yes

4. Have the authors made all data underlying the findings in their manuscript fully available?

Reviewer #2: Yes

5. Is the manuscript presented in an intelligible fashion and written in standard English?

Reviewer #2: No

Reviewer #2: The paper « Cost-effective DNA extraction method optimized for high yield and long fragments from coastal sediments » proposes a new « home-made » DNA extraction method efficient for sediments, particarly for sandy or low-biomass sediments. It is the second article submission after review. Authors noted seriously all remarks from my reviews and answered in detail on my questions.

Comments on 1 :

Thank you for your response. It is clearer.

Comments on 2a and 5 :

Many thanks for your tests. But it is still not clear. I suggested in my review to do some amplifcation tests on 18S-V7, but in your new submission, you studied both 18S-V7 (eukaryotes) and 16S-V4V5 (prokaryotes). Which marker do you finally amplified with the Phusion ? 18S-V7 or 16S-V4V5 ?

Comments on 2b :

Many thanks for your explanations.

Comments on 3 :

Thank you very much for your answer. I am disappointed to not have technical replicates in the PCR. We know that amplification step adds a lot of biaises for metabarcoding studies, even at 25 cycles, that’s why technical replicates (at least 3) are recommended, specially in sediments. Maybe you had to face resource constraints, but it is a critical point.

Comments on 4 :

Thank you very much for your explanations and I totally understand your resource constraints. Nevertheless, this LM DNA extraction protocol is promoted for getting high-molecular-weight DNA. It would have been interesting to show some results on ONT or PacBio.

I still have new questions about the modified article :

Line180-182 for 18S-V7 failed amplifications, you mentionned you reamplifed with 20 ng of DNA.

Did you test with the PCR with less DNA amounts (1 or 0.1 ng startin material) ? Despite the fact DNA extraction methods have a lot of purification steps, there are still a lot of inhibitors in DNA extracts from sediments. If you dilute the DNA input in the PCR, you also dilute PCR inhibitors and increase the amplification yield.

S2_File : it is not « furnisher », but « Supplier ». And Fisher brand is the same than Thermofisher.

**Do you want your identity to be public for this peer review?** For information about this choice, including consent withdrawal, please see our Privacy Policy

Reviewer #2: No

---

## [Author Response · Author response to Decision Letter 2]

6 Feb 2026

Comments on 2a and 5:

Many thanks for your tests. But it is still not clear. I suggested in my review to do some amplifcation tests on 18S-V7, but in your new submission, you studied both 18S-V7 (eukaryotes) and 16S-V4V5 (prokaryotes). Which marker do you finally amplified with the Phusion ? 18S-V7 or 16S-V4V5 ?

Author response: Thank you for your comment and we apologize for the lack of clarity. In the original manuscript, both 18S-V7 and 16S-V4V5 were amplified using the Promega polymerase. Following your recommendation during the first round of review, we tested the 18S-V7 marker with the Phusion polymerase, as Promega may not be optimal for marine microbes, especially protists. We apologize for not clearly stating that these amplification tests were performed only on the 18S-V7 marker. These tests resulted in a lower amplification success with Phusion compared to Promega. As a result, we retained the Promega polymerase for all amplifications presented in the manuscript, for both 18S-V7 and 16S-V4V5.

Comments on 3:

Thank you very much for your answer. I am disappointed to not have technical replicates in the PCR. We know that amplification step adds a lot of biaises for metabarcoding studies, even at 25 cycles, that’s why technical replicates (at least 3) are recommended, specially in sediments. Maybe you had to face resource constraints, but it is a critical point.

Author response: Thank you for your comment. We agree that PCR technical replicates are recommended in metabarcoding studies to limit amplification biases. In this study, we prioritized experimental replicates, which we consider to capture a broader source of variability than PCR-induced errors alone. We acknowledged this limitation in the manuscript (line 190) and will implement PCR technical replicates in future work.

Comments on 4:

Thank you very much for your explanations and I totally understand your resource constraints. Nevertheless, this LM DNA extraction protocol is promoted for getting high-molecular-weight DNA. It would have been interesting to show some results on ONT or PacBio.

Author response: Thank you for your comment. While we acknowledge that ONT or PacBio sequencing can provide higher taxonomic resolution, the primary aim of this study was not to optimize taxonomic precision but to compare community-level patterns and method-specific differences across extraction protocols. In this context, previous work has shown that alpha-diversity patterns are consistent between short- and long-read metabarcoding approaches (Mittelstrass et al. 2025, https://doi.org/10.1186/s40793-025-00712-7 ; Doorenspleet et al. 2025 https://doi.org/10.7717/peerj.19158). Similar results were found comparing beta-diversity patterns (Doorenspleet et al. 2025; Bludeau et al. 2024 https://doi.org/10.1002/edn3.70084). Consequently, we do not expect the use of long-read sequencing to substantially alter the main conclusions of this study, and we therefore consider the current approach appropriate to address our research objectives.

I still have new questions about the modified article :

Line180-182 for 18S-V7 failed amplifications, you mentionned you reamplifed with 20 ng of DNA.

Did you test with the PCR with less DNA amounts (1 or 0.1 ng startin material) ? Despite the fact DNA extraction methods have a lot of purification steps, there are still a lot of inhibitors in DNA extracts from sediments. If you dilute the DNA input in the PCR, you also dilute PCR inhibitors and increase the amplification yield.

Author response: Thank you for raising this point. We agree that diluting DNA input can in some cases improve PCR amplification by reducing the concentration of inhibitors, particularly in sediment-derived DNA extracts. In this study, samples that failed to amplify with the 18S-V7 marker successfully amplified with the 16S-V4V5 marker, suggesting that PCR inhibition was unlikely. Instead, these failures were more consistent with a low amount of target DNA, possibly related to the low eukaryotic biomass commonly observed in sediment samples. This interpretation is supported by the improved amplification success of the 18S-V7 marker using more input DNA (20 ng instead of 10 ng). In addition, preliminary tests performed on similar sediment types using lower DNA inputs (1 ng) did not improve amplification success for the 18S-V7 marker. These results suggest that, in our case, amplification failure was more likely related to low target DNA abundance rather than PCR inhibition.

S2_File : it is not « furnisher », but « Supplier ». And Fisher brand is the same than Thermofisher.

Author response: Thank you for this correction. We replaced “Furnisher” with “Supplier” and corrected “Fisher” for “ThermoFisher” in S2_File.

---

## [Editor Report · Decision Letter 2]

11 Feb 2026

Cost-effective DNA extraction method optimized for high yield and long fragments from coastal sediments

PONE-D-25-39169R2

Dear Dr. Navarro,

We’re pleased to inform you that your manuscript has been judged scientifically suitable for publication and will be formally accepted for publication once it meets all outstanding technical requirements.

Kind regards,

Raffaella Casotti

Academic Editor

PLOS One

Additional Editor Comments (optional):

Thank you for clarifications
---

## [Editor Report · Acceptance letter]

PONE-D-25-39169R2

PLOS One

Dear Dr. Navarro,

I'm pleased to inform you that your manuscript has been deemed suitable for publication in PLOS One. Congratulations! Your manuscript is now being handed over to our production team.

Kind regards,

on behalf of

Dr. Raffaella Casotti

Academic Editor

PLOS One